# Evaluation of the effectiveness of the physician education program on primary palliative care in heart failure

Tatsuhiro Shibata[1], Shogo Oishi[2], Atsushi Mizuno[3,4,5,6], Takashi Ohmori[7], Tomonao Okamura[7], Hideyuki Kashiwagi[7], Akihiro Sakashita[2,8], Takuya Kishi[9], Hitoshi Obara[10], Tatsuyuki Kakuma[10], Yoshihiro Fukumoto[1]*

1 Division of Cardiovascular Medicine, Department of Internal Medicine, Kurume University School of Medicine, Fukuoka, Japan, 2 Department of Cardiology, Himeji Cardiovascular Center, Hyogo, Japan, 3 Department of Cardiovascular Medicine, St. Luke's International Hospital, Tokyo, Japan, 4 Department of Cardiovascular Medicine, Juntendo University Graduate School of Medicine, Tokyo, Japan, 5 Penn Medicine Nudge Unit, University of Pennsylvania, Philadelphia, PA, United States of America, 6 Leonard Davis Institute for Health Economics, University of Pennsylvania, Philadelphia, PA, United States of America, 7 Department of Transitional and Palliative Care, Iizuka Hospital, Fukuoka, Japan, 8 Department of Palliative Medicine, Kobe University School of Medicine, Hyogo, Japan, 9 Faculty of Health and Welfare Sciences in Fukuoka, International University of Health and Welfare, Fukuoka, Japan, 10 Biostatistics Center, Kurume University School of Medicine, Kurume, Japan

* fukumoto_yoshihiro@med.kurume-u.ac.jp

**Data Availability Statement:** All relevant data are within the paper and its Supporting Information files.

## Abstract

Major cardiology societies' guidelines support integrating palliative care into heart failure (HF) care. This study aimed to identify the effectiveness of the HEart failure Palliative care Training program for comprehensive care providers (HEPT), a physician education program on primary palliative care in HF. We performed a pre- and post-test survey to evaluate HEPT outcomes. Physician-reported practices, difficulties and knowledge were evaluated using the Palliative Care Self-Reported Practices Scale in HF (PCPS-HF), Palliative Care Difficulties Scale in HF (PCDS-HF), and Palliative care knowledge Test in HF (PT-HF), respectively. Structural equation models (SEM) were used to estimate path coefficients for PCPS-HF, PCDS-HF, and PT-HF. A total of 207 physicians participated in the HEPT between February 2018 and July 2019, and 148 questionnaires were ultimately analyzed. The total PCPS-HF, PCDS-HF, and PT-HF scores were significantly improved 6 months after HEPT completion (61.1 vs 67.7, p<0.001, 54.9 vs 45.1, p<0.001, and 20.8 vs 25.7, p<0.001, respectively). SEM analysis showed that for pre-post difference (Dif) PCPS-HF, "clinical experience of more than 14 years" and pre-test score had significant negative effects (-2.31, p = 0.048, 0.52, p<0.001, respectively). For Dif PCDS-HF, ≥ "28 years old or older" had a significant positive direct effect (13.63, p<0.001), although the pre-test score had a negative direct effect (-0.56, p<0.001). For PT-HF, "involvement in more than 50 HF patients' treatment in the past year" showed a positive direct effect (0.72, p = 0.046), although the pre-test score showed a negative effect (-0.78, p<0.001). Physicians who completed the HEPT showed significant improvements in practice, difficulty, and knowledge scales in HF palliative care.

**Funding:** This research was supported by grants from the Sasakawa Health Foundation (T.S.), the Cardiovascular Research Fund (T.S.), and AMED under Grant Number JP18ek0210072 (A.M.), Japan. The funders had no role in study design, data collection, and analysis, decision to publish, and preparation of the manuscript.

**Competing interests:** The authors have declared that no competing interests exist.

**Abbreviations:** ACC, American College of Cardiology; ACP, advance care planning; CART, classification and regressing tree model; COCATS, Core Cardiovascular Training Statement; Dif, difference; HEPT, Heart failure Palliative care Training program for comprehensive care providers; HF, heart failure; ICD, implantable cardioverter defibrillators; MCS, mechanical circulatory support; PCDS, Palliative Care Difficulties Scale; PCDS-HF, Palliative Care Difficulties Scale in heart failure; PCPS, Palliative Care Self-Reported Practices Scale; PCPS-HF, Palliative Care Self-Reported Practices Scale in heart failure; PEACE, Palliative care Emphasis program on symptom management and Assessment for Continuous medical Education; PT-HF, Palliative care knowledge Test in heart failure; QOL, quality of life; SEM, Structural equation models.

## Introduction

During the present decade, the increase in patients with heart failure (HF) has become an important healthcare issue worldwide. Palliative care is a multidisciplinary healthcare approach that focuses on optimizing quality of life (QOL) and alleviating the suffering of patients and families living with serious illnesses, regardless of their prognosis [1]. Although most evidence of palliative care comes from oncology, several recent reports have suggested that palliative care interventions for HF patients can improve symptom burden and QOL [2–5]. These trends have led to major HF guidelines supporting the integration of palliative care into HF care [6, 7].

In contrast, it is difficult for only a limited number of palliative care professionals to provide all levels of palliative care because the role of modern palliative care has expanded beyond the end of life and includes not only cancer but also the early stages of any life-threatening illness [8]. Therefore, we need a system that divides palliative care into primary palliative care, which can be provided by all clinicians, and specialized palliative care, which can be provided by specialists for more complex and challenging issues. This would ensure that appropriate care is provided to all patients who need palliative care [8, 9]. In cancer care, the Cancer Control Act of Japan, approved in 2006, states that palliative care should be provided at the time of cancer diagnosis and requires all physicians engaged in cancer treatment to attend a postgraduate education program on primary palliative care. The program is called the Palliative care Emphasis program on symptom management and Assessment for Continuous medical Education (PEACE), and its effectiveness has been shown in previous reports [10, 11]. However, a nationwide survey of Japanese Circulation Society-authorized cardiology training hospitals indicated that most cardiologists had received little or no education on palliative care [12]. Moreover, there are disease-specific challenges, such as the illness trajectory and disease management in HF, which are different from cancer, including implantable cardioverter defibrillators (ICD) and mechanical circulatory support (MCS) at the end of life [13]. Despite this program's success, there has been no primary palliative care training program tailored to HF clinicians worldwide.

In October 2017, therefore, we began developing a primary palliative care educational program targeted toward all physicians engaged in HF care. This educational program, called the HEart failure Palliative care Training program for comprehensive care providers (HEPT), is a 325-minute one-day program developed by the authors based on available evidence and expert opinions on primary palliative care in HF [8, 14–19].

The purpose of this study was to identify the effectiveness of a physician education program on primary palliative care in HF by examining changes in physician-reported practices, difficulties, and knowledge due to participation in the HEPT. In addition to directly testing significant changes in the scores, the effects of the participants' characteristics and pre-test score on score changes were examined simultaneously using structural equation models (SEM) [20].

## Methods

HEPT consists of six modules that combine interactive didactic lectures and small-group sessions (Table 1). In this study, we performed a pre- and post-test survey to evaluate HEPT outcomes. Scores in the pre- and post-test were compared for each participant to determine whether there were any changes in physician-reported practices or in difficulties with and knowledge of palliative care in HF. We modified a palliative care assessment tool which has already been validated in oncology to be suitable for HF in order to assess practices and difficulties. We also developed a new palliative care knowledge test in HF to assess participants' knowledge. The pre-test was conducted just before the start of the HEPT program. The post-

**Table 1. Outline of the HEPT content.**

| Module | Title | Educational style | Time, min |
|---|---|---|---|
| 1 | **Guidance on the outline of this workshop**<br>• Overview of curriculum and organizing framework | | 10 |
| 2 | **Overview of palliative care for heart failure patients**<br>• Definition of PC<br>• Needs and current status of PC for patients with HF<br>• Stress and suffering over the course of the HF experience<br>• Similarities and differences between PC for cancer and HF<br>• Concept of PC intervention provided alongside cardiologic management | Interactive-didactic lecture | 45 |
| 3 | **Decision making and advance care planning in heart failure**<br>• Definition of ACP<br>• Difference between ACP and advance directive<br>• Trigger for the consideration of ACP<br>• Essential component of ACP<br>• Communication skills | Interactive-didactic lecture and small-group session | 90 |
| 4 | **Refractory symptom management in heart failure**<br>• Systematic approach to symptom assessment<br>• Appropriate use of opioids and other medication for management of refractory symptoms<br>• Non-pharmacotherapy for refractory symptoms | Interactive-didactic lecture | 45 |
| 5 | **Psychosocial problems in heart failure**<br>• Screening and assessment for depression<br>• Effective management of depression<br>• Strategies for the prevention and treatment of delirium | Interactive-didactic lecture | 45 |
| 6 | **Ethical issues in heart failure**<br>• Principles of clinical ethics<br>• Ethical issues in heart failure (e.g., Do-not-resuscitate order and ICD deactivation) | Interactive-didactic lecture and small-group session | 90 |
| | | Total | 325 |

HEPT; HEart failure Palliative care Training program for comprehensive care provider, PC; palliative care, HF; heart failure, ACP; advance care planning, ICD; implantable cardioverter defibrillators.

test was conducted using a mailed questionnaire six months after the completion of the HEPT. The pre-test and six-month post-test questionnaires contained the same content. To protect confidentiality and to match the pre- and post-test data, each participant was identified by their unique identifier number written on the pre- and post-test questionnaires. This study was approved by the institutional review board of Kurume University (No. 18067) and conducted in accordance with the Declaration of Helsinki. All the participants gave written informed consent.

## Subjects

This study included all physicians who participated in one of seven HEPT sessions held in six regions (Kurume, Fukuoka, Hiroshima, Kobe, Tokyo, and Sendai) in Japan between February 2018 and July 2019. Physicians wishing to participate in a HEPT session were recruited through a website (http://hept.main.jp/). Participants were informed by the researchers that their participation in this study was voluntary. The researcher distributed an informed consent form to each participant before HEPT to allow them to consider if they would participate in the study. Physicians could participate in the HEPT even if they did not consent to participate in the study.

## Participant characteristics

Age, gender, years of clinical experience, specialty, and workplace were recorded. We also recorded clinical experience (working in a palliative care unit and the number of HF patients

treated and opioids prescribed in the past year), experience with end-stage HF care in the past year, and previous attendance at the PEACE.

## Measurements

Physician-reported practices in HF palliative care were measured using the Palliative Care Self-Reported Practices Scale [21], modified for HF (PCPS-HF; S1 Table). The original PCPS consisted of 18 items on six subscales (pain, dyspnea, delirium, dying-phase care, communication, and patient- and family-centered care). In PCPS-HF, 17 items except for the item on the dose of the rescue opioid from original PCPS were chosen. In these 17 items, the word "pain" was changed to "symptom". Each item was evaluated using a Likert-type scale ranging from 1 (never) to 5 (always). The PCPS-HF scores ranged from 17 to 85, with a higher score indicating a higher level of performance in the recommended practices.

Physician-reported difficulties with providing palliative care in HF were measured using the Palliative Care Difficulties Scale [21], modified for HF (PCDS-HF; S2 Table). The original PCDS consisted of 15 items on five subscales (alleviation of symptoms, expert support, communication in multidisciplinary teams, communication with patient and family, and community coordination). PCDS-HF added the item "involvement of palliative care" to the original PCDS, making it a 16-item scale consisting of 6 subscales. Additionally, some words were changed in the following manner: "cancer pain" to "symptoms" and "cancer patients" to "heart failure patients." Each item was evaluated using a Likert-type scale ranging from 1 (never) to 5 (always). The PCDS-HF scores ranged from 16 to 80, with a lower score indicating fewer perceived difficulty.

Physician-reported knowledge of palliative care in HF was measured by the authors' proposed Palliative care knowledge Test in HF (PT-HF; S3 Table). This test was a 29-item questionnaire with a single correct answer that tested physicians' knowledge of the philosophy of palliative care, decision making and advance care planning (ACP), refractory symptom management, psychosocial problems, and clinical ethics in HF, answerable by "true," "false," and "don't know." The score consisted of the arithmetic sum of all correct items (with a maximum score of 29), suggesting that a higher PT-HF score indicated greater knowledge.

## Statistical analysis

All continuous variables were shown as mean (standard deviation, SD) or median (interquartile range, IQR), as appropriate. A nonparametric Wilcoxon signed rank test was used to assess the difference between pre- and post-test scores for participants' practices, difficulties, and knowledge of primary palliative care in HF. The following sequential data analysis steps were performed to construct the path diagrams shown in Figs 1–3. First, the full path diagrams were specified, where the score changes and the pre-test scores were considered as endogenous variables, and the other variables shown in Table 2 were treated as exogenous variables. Before the full SEM model was fitted, all endogenous variables were dichotomized to decrease the number of parameters in the model and to enhance the interpretability of the parameter estimates. Dichotomization was carried out in an exploratory manner by inspecting both the frequency distributions and the magnitude of parameter estimates. To aid the dichotomizing process, the classification and regressing tree model (CART) was also employed, where the score change was used as the response variable, and each exogenous variable was an explanatory variable. To this end, age was classified into $\geq$28 and <28 years, years of clinical experience into $\geq$14 and <14 years, clinical experience in treating HF patients in the past year into $\geq$50 and <50 patients, clinical experience in treating terminally ill HF patients in the past year into $\geq$10 and <10, and clinical experience in prescribing opioids in the past year into $\geq$10 and <10. Next, a

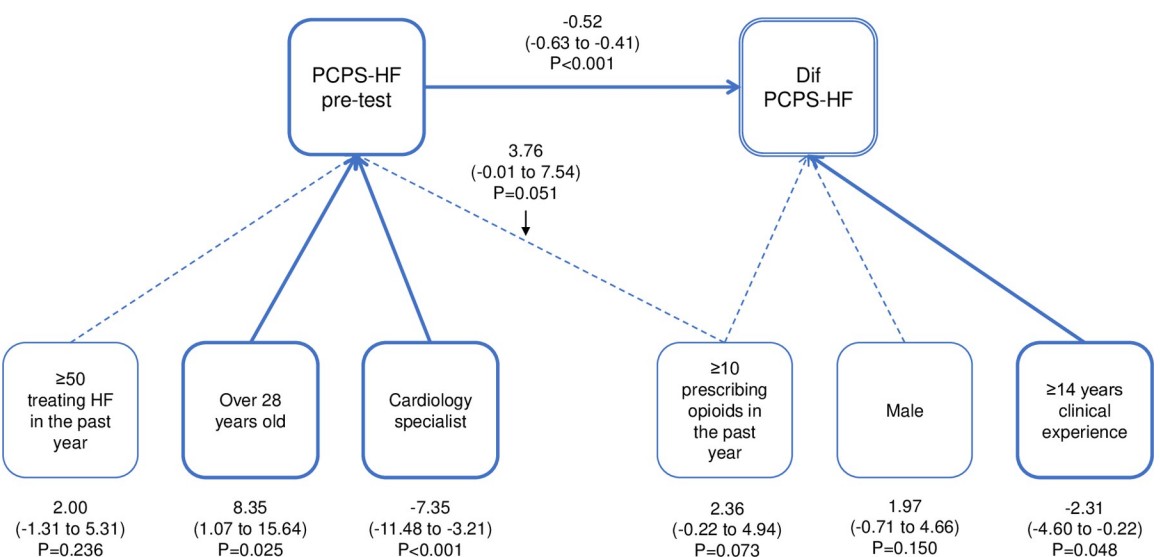

**Fig 1. Path coefficient diagram of PCPS-HF.** PCPS-HF indicates the Palliative Care Self-Reported Practices Scale modified for heart failure, and HF indicates heart failure. Data are expressed as a coefficient (95% confidence interval) and P value.

reduced path diagram was created based on the criteria in which a path would be deleted when the p-value for the corresponding path coefficient was greater than 0.2. The final SEM model was obtained by fitting a reduced-path diagram. All p values <0.05 were considered statistically significant. All analyses were performed using JMP Pro 14 (SAS Institute Inc., Cary, NC, USA) and STATA/MP 16.1 (StataCorp LLC, College Station, TX, USA).

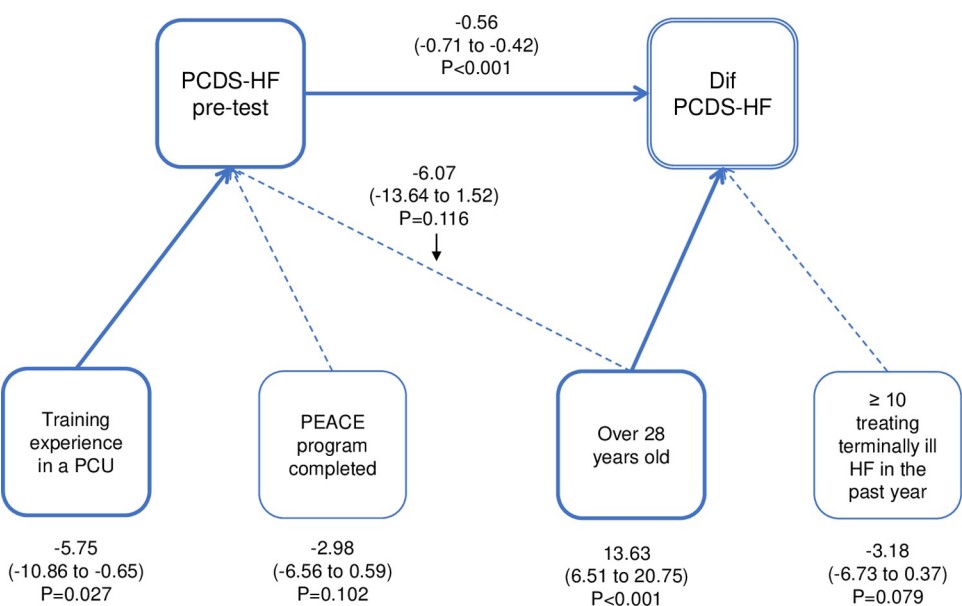

**Fig 2. Path coefficient diagram of PCDS-HF.** PCDS-HF, Palliative Care Difficulties Scale modified for HF; PCU, palliative care unit; PEACE, palliative care emphasis program on symptom management and assessment for continuous medical education, and HF indicates heart failure. Data are expressed as a coefficient (95% confidence interval) and P value.

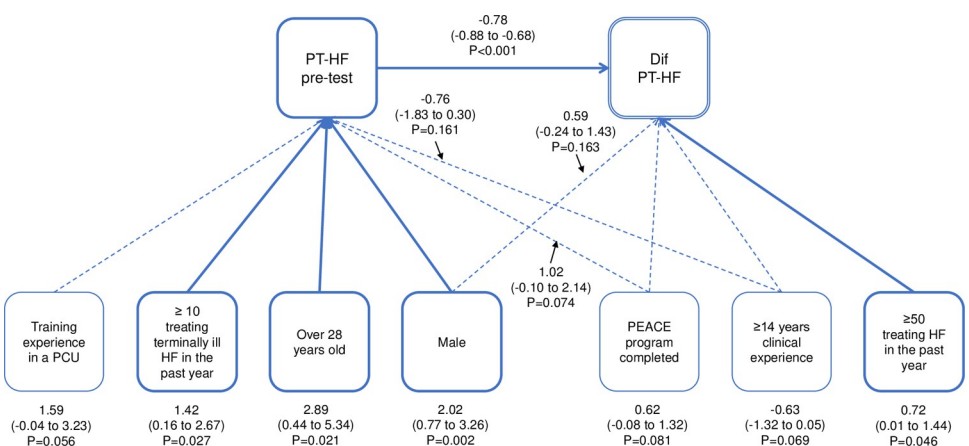

**Fig 3. Path coefficient diagram of PT-HF.** PT-HF, palliative care knowledge test in HF; PCU, palliative care unit; HF, heart failure; and PEACE, Palliative care Emphasis program on symptom management, and Assessment for Continuous medical Education. Data are expressed as a coefficient (95% confidence interval) and P value.

## Results

A total of 207 physicians participated in the study. Table 2 shows the baseline characteristics of the study patients. Of the participating physicians, 150 (72%) returned for the six-month post-test questionnaires, and two responses were excluded from the analyses due to the lack of data, so a total of 148 questionnaires were analyzed in this study. Table 3 details the changes in PCPS-HF, PCDS-HF, and PT-HF scores before and six months after completion of the HEPT.

**Table 2. Baseline characteristics of participants.**

|  | (n = 207) N (%) |
|---|---|
| Age, y | 37.2 ± 8.0 |
| Male | 162 (78.3) |
| Years of clinical experiences, y | 11.9 ± 7.6 |
| Specialty |  |
| Cardiology | 159 (76.8) |
| Palliative care | 14 (6.8) |
| Primary care | 30 (14.5) |
| Others | 4 (1.9) |
| Workplace |  |
| Designated cancer hospitals | 119 (57.5) |
| Hospital with over 200 beds | 64 (30.9) |
| Hospital with under 199 beds | 17 (8.2) |
| Clinic | 6 (2.9) |
| Others | 1 (0.5) |
| Clinical experience in treating heart failure patients in the past year |  |
| None | 3 (1.4) |
| 1–9 | 28 (13.5) |
| 10–49 | 98 (47.3) |
| 50–99 | 42 (20.3) |
| ≥100 | 36 (17.4) |
| Clinical experience in treating terminally ill heart failure patients in the past year |  |

*(Continued)*

**Table 2.** (Continued)

| | (n = 207) N (%) |
|---|---|
| None | 11 (5.3) |
| 1–9 | 146 (70.5) |
| 10–49 | 43 (20.8) |
| 50–99 | 6 (2.9) |
| ≥100 | 1 (0.5) |
| Clinical experience in prescribing opioids in the past year | |
| None | 24 (11.6) |
| 1–9 | 129 (62.3) |
| 10–49 | 34 (16.4) |
| 50–99 | 14 (6.8) |
| ≥100 | 6 (2.9) |
| Training experiences in a palliative care unit | |
| Yes | 6 (2.9) |
| No | 201 (97.1) |
| Completed the primary palliative care (PEACE) program | |
| Not taken | 136 (65.7) |
| Completed | 71 (34.3) |

Data are expressed as mean±SD or n (%). PEACE, Palliative care Emphasis program on symptom management and Assessment for Continuous medical Education.

Compared to baseline, physician-reported practices in HF palliative care six months after completion of the HEPT significantly improved the total PCPS-HF scores of 62 (IQR 55–68) and 67 (IQR 62–74), respectively (p<0.001, **Table 3**). A significant increase in scores was observed for all subscales of the PCPS-HF. Regarding physician-reported difficulties in HF palliative care, the total PCDS-HF score was significantly lower six months after the completion of the HEPT than at baseline (56 vs. 45, p<0.001, **Table 3**). All subscales of the PCDS-HF showed significant improvement, except for the item "involvement of palliative care." Furthermore, there was a significant increase in the level of palliative care knowledge measured by total PT-HF scores six months after compared to before the HEPT (21 vs. 26, p<0.001, **Table 3**).

Using structural equation modeling (SEM), path coefficients were estimated for PCPS-HF, PCDS-HF, and PT-HF. Fig 1 shows the path diagram of PCPS-HF. More than 14 years of clinical experience and PCPS-HF pre-test score showed a significantly negative effect on the pre-post difference score denoted as "Dif PCPS-HF" (-2.31, p = 0.048, -0.52, p<0.001, respectively). "Age 28 years or older" showed a significant effect on the pre-test score (8.35, p = 0.025). As the pre-test score is negatively associated with Dif PCPS-HF, age >28years has an indirect negative association with Dif PCPS-HF. Similarly, cardiology specialty showed an indirect positive effect (-7.35, p<0.001). In the path diagram of PCDS-HF (Fig 2), "age 28 years or older" showed a significant positive direct effect (13.63, p<0.001), while the PCDS-HF pre-test showed a negative direct effect (-0.56, p<0.001) against Dif PCDS-HF. Indirectly, training experiences in a palliative care unit showed a positive effect (-5.75, p = 0.027), mediated by a negative PCDS-HF pre-test score. In the path diagram of PT-HF shown in Fig 3, more than 50 clinical experiences in treating HF patients in the past year showed a positive direct effect (0.72, p = 0.046), and the PT-HF pre-test resulted in a negative association (-0.78, p<0.001) for Dif PT-HF. Male (2.02, p = 0.002), older than 28 years (2.89. p = 0.021), and more than 10 clinical experiences in treating terminally ill HF patients in the past year (1.42,

**Table 3. Change in PCPS-HF, PCDS-HF and PT-HF for each domain.**

| | Before HEPT | 6 months after HEPT | P value |
|---|---|---|---|
| PCPS in heart failure (PCPS-HF)* | | | |
| Total (score range, 17–85) | 62 (55–68) | 67 (62–74) | <0.001 |
| Symptom evaluation (score range, 2–10) | 7 (5–8) | 8 (7–9) | <0.001 |
| Dyspnea (score range, 3–15) | 11 (8–12) | 12 (10–13) | <0.001 |
| Delirium (score range, 3–15) | 9 (7–11) | 10 (9–12) | <0.001 |
| Dying-phase care (score range, 3–15) | 11 (9–12) | 12 (10–14) | <0.001 |
| Communication (score range, 3–15) | 12 (11–14) | 14 (12–15) | <0.001 |
| Patient- and family-centered care (score range, 3–15) | 12 (10–14) | 13 (12–15) | <0.001 |
| PCDS in heart failure (PCDS-HF)† | | | |
| Total (score range, 16–80) | 56 (48–62) | 45 (38–52) | <0.001 |
| Involvement of palliative care (score range, 1–5) | 4 (3–5) | 4 (3–4) | 0.140 |
| Alleviating symptoms (score range, 3–15) | 12 (10–13) | 8 (6–10) | <0.001 |
| Expert support (score range, 3–15) | 10 (7–12) | 7 (5–11) | 0.002 |
| Communication in multidisciplinary teams (score range, 3–15) | 10 (8–12) | 8 (6–9) | <0.001 |
| Communication with patient and family (score range, 3–15) | 10 (9–12) | 9 (6–10) | <0.001 |
| Community coordination (score range, 3–15) | 11 (9–13) | 9 (6–11) | <0.001 |
| Palliative care knowledge test in heart failure (PT-HF)‡ | | | |
| Total (score range, 0–29) | 21 (18–24) | 26 (24–28) | <0.001 |
| Philosophy of palliative care in heart failure (score range, 0–6) | 5 (4–5) | 5 (5–6) | <0.001 |
| Decision making and advance care planning in heart failure (score range, 0–5) | 4 (3–5) | 5 (5–5) | <0.001 |
| Refractory symptom management in heart failure (score range, 0–6) | 4 (4–5) | 6 (5–6) | <0.001 |
| Psychosocial problems in heart failure (score range, 0–6) | 4 (4–5) | 6 (5–6) | <0.001 |
| Clinical ethics in heart failure (score range, 0–6) | 3 (2–4) | 5 (5–6) | <0.001 |

Data are expressed as median (interquartile range 25–75%). *Higher score indicates higher level of performance of recommended practices. †Higher score indicates more difficulties perceived. ‡Higher score indicates more accurate knowledge.

p = 0.027) were all negatively associated with Dif PT-HF through the pre-test score. A history of participation in PEACE did not show significance in any of the scores.

## Discussion

### Main findings

The present study indicated that measures of physician-reported practice, difficulty, and knowledge scales in HF palliative care significantly improved six months after completion of the HEPT, an education program focused on primary palliative care for HF. SEM analysis showed that for Dif PCPS-HF, more than 14 years of clinical experience and PCPS-HF pre-test score had a significant negative effect, while for Dif PCDS-HF, 28 years of age or older had a significant positive direct effect, but the PCDS-HF pre-test had a negative direct effect. Moreover, for PT-HF, being involved in the treatment of more than 50 HF patients in the past year showed a positive direct effect, but the PT-HF pre-test showed a negative effect. To the best of our knowledge, this is the first study to examine the effectiveness of an HF-specific primary palliative care education program for physicians.

The current study did not find any improvement for the new subscale of "involvement of palliative care" added to the PCDS-HF. Integrating palliative care into HF practice is challenging. There is often a misconception among patients, their families, and non-palliative physicians that palliative care is relevant only at the end of life, and this misconception is a major

barrier to HF patients' access to palliative care. However, HF patients often require a holistic and multidisciplinary approach throughout the course of their disease and not simply at the end of life. They must manage physical and psychosocial problems, have an understanding of their disease, and receive support for the ACP process to ensure appropriate treatment based on the patient's goals and values. HF guidelines emphasize that palliative care should be introduced early in the course of the disease [6, 7, 22].

In the present study, we proposed a new PT-HF because there has been no measurement approach to assess knowledge of HF primary palliative care. Crousillat et al. [23] defined essential palliative care competencies for cardiology fellows based on the American College of Cardiology's (ACC) 2015 Core Cardiovascular Training Statement (COCATS 4) and key guidelines. Recently, two quality indicators of palliative care for cardiovascular disease using the Delphi method have been developed in Japan [13, 24]. Most of these competencies and indicators have been included in the contents of the HEPT and PT-HF, except competency for hospice indications as hospices are not currently available to HF patients in Japan. The PT-HF needs to be validated in future studies.

Because of the limited availability of specialized palliative care providers, standardizing HF primary palliative competencies and providing appropriate educational opportunities for all physicians involved in HF care is necessary to ensure access to palliative care for all HF patients [23]. It is also important to learn when it is appropriate to refer patients to palliative care specialists, such as for intractable symptom management or complex decision-making (e.g., disagreements in goals between patient and family, unrealistic expectations of treatment). However, palliative care is rarely included in undergraduate medical curricula, and is not included in the competency components of the current training curriculum for cardiologists in Japan. It should be included in pre- and post-graduate education in cardiology in near future. The results of the SEM analysis in this study showed that the pre-test score had a direct negative effect on the Dif of each scale. In addition, the length of clinical experience had a negative effect on Dif PCPS-HF. These results suggest that the main target population may be young residents with limited clinical experience and limited knowledge and practice in palliative care. However, being older than 28 years had a positive impact on Dif PCDS-HF, and having a relatively high number of HF patients treated in the past year had a positive impact on Dif PT-HF. Moreover, the cardiology specialty had a positive indirect effect on Dif PCPS-HF, and experience in a palliative care unit had a positive indirect effect on Dif PCDS-HF. These results suggest that the HEPT may be useful not only for young residents but also for physicians with extensive clinical experience in HF and palliative care practice. Although the skills required for primary palliative care are often cross-disease, it is not always appropriate to assume that the palliative care framework used for cancer patients is optimal for patients with chronic non-malignant illnesses such as HF [16]. There are disease-specific issues to be understood, such as the unpredictable trajectory of HF, the ambiguity regarding the differences between therapeutic HF treatment and palliative care, and the management of ICD and MCS at the end of life [13]. In the clinical settings, the number of ICD implantations is increasing to prevent sudden death; however, shock therapy may be repeated at the end of life, resulting in patient distress, poor quality of death, and family distress. Currently, only a limited number of physicians have experience with ICD deactivation [12]. We hope that the HEPT participants will discuss about ICD deactivation and lead to a wider dissemination of this concept. We believe that HEPT is an efficient and valuable short-term program to learn these elements. Although this study was conducted for physicians, non-physician medical staff, such as heart failure nurses, are also deeply involved in palliative care in clinical practice. It is expected that a training system similar to HEPT will be established for medical staff.

## Limitations

This study had several limitations. First, it is unclear whether the improvements in physician-reported measures reflected the actual quality of palliative care for patients with HF. Further research investigating the impact on QOL and satisfaction of HF patients and their families will be needed to assess the true outcomes of primary palliative care education as our next study. Further, a randomized trial should be considered to evaluate the patients' and/or families' satisfaction. Second, there may be a response bias. However, because the response rate for the previous follow-up survey to the PEACE was 38.1% [10], the present survey had a relatively high response rate (72%) for a physician-based survey. Therefore, a more reliable follow-up system needs to be established. Third, selection bias may have affected the results. Physicians who participated in this study had a strong interest in palliative care for HF. They were relatively young, and 34.3% had a history of participation in PEACE. Further research is needed to determine whether the results apply to all types of physicians caring for patients with HF, and also to evaluate additionally if HEPT is widely adopted in future. Fourth, we used the tool, which has changed from "pain" to "symptom" of the cancer PCPS and PCDS list for heart failure. Currently, there is no tool to evaluate the effectiveness of education on HF palliative care. Therefore, we should validate this tool after data accumulation.

## Conclusion

With the increased attention to HF palliative care, there is a need for appropriate educational opportunities in this practice. Physicians who completed the HEPT significantly improved their scores on the practice, difficulty, and knowledge scales in HF palliative care. The HEPT may increase the number of physicians with primary palliative skills in HF, thereby providing normalized, seamless, and long-term palliative care throughout the HF experience, not only at the end of life.

## Supporting information

**S1 Table. The Palliative Care Self-reported Practices Scale in heart failure (PCPS-HF).**
(XLSX)

**S2 Table. The Palliative Care Difficulties Scale in heart failure (PCDS-HF).**
(XLSX)

**S3 Table. Palliative care knowledge Test in heart failure (PT-HF).**
(XLSX)

## Acknowledgments

The authors are grateful for the contributions of Yasuki Kihara, Hiroyuki Tsutsui, and observers of the HEPT (Kenichi Ichita, Toshihisa Anzai, Yoshiyuki Kizawa, Ryuichi Sekine, Ryo Yamamoto, Tomomi Sano, Naoki Horikawa, Saori Yamabe, Toshimi Ikegami, Kyoko Nuruki, and Mika Sumiyoshi).

## Author Contributions

**Conceptualization:** Tatsuhiro Shibata, Shogo Oishi, Atsushi Mizuno, Takashi Ohmori, Tomonao Okamura, Hideyuki Kashiwagi, Akihiro Sakashita, Takuya Kishi.

**Data curation:** Tatsuhiro Shibata, Shogo Oishi, Atsushi Mizuno, Takashi Ohmori, Tomonao Okamura, Hideyuki Kashiwagi, Akihiro Sakashita, Takuya Kishi.

**Formal analysis:** Tatsuhiro Shibata.

**Funding acquisition:** Tatsuhiro Shibata.

**Investigation:** Tatsuhiro Shibata.

**Methodology:** Hitoshi Obara, Tatsuyuki Kakuma.

**Project administration:** Tatsuhiro Shibata, Shogo Oishi.

**Supervision:** Tatsuyuki Kakuma, Yoshihiro Fukumoto.

**Visualization:** Hitoshi Obara.

**Writing – original draft:** Tatsuhiro Shibata.

**Writing – review & editing:** Yoshihiro Fukumoto.

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
