## [Decision Letter · Decision Letter 0]

14 Dec 2021

PONE-D-21-27736Evaluation of the effectiveness of the physician education program on primary palliative care in heart failurePLOS ONE

Dear Dr. Fukumoto,

Thank you for submitting your manuscript to PLOS ONE. After careful consideration, we feel that it has merit but does not fully meet PLOS ONE’s publication criteria as it currently stands. Therefore, we invite you to submit a revised version of the manuscript that addresses the points raised during the review process.

We look forward to receiving your revised manuscript.

Kind regards,

Alexander H. Maass, M.D., Ph.D.

Academic Editor

PLOS ONE

a) Did participants provide their written or verbal informed consent to participate in this study?

“This research was supported by grants from the Sasakawa Health Foundation (T.S.), the Cardiovascular Research Fund (T.S.), and AMED under Grant Number JP18ek0210072 (A.M.), Japan.”

“The author(s) declared no potential conflicts of interest with respect to the research, authorship, and/or publication of this article.”

Reviewers' comments:

Reviewer's Responses to Questions

**Comments to the Author**

1. Is the manuscript technically sound, and do the data support the conclusions?

Reviewer #1: Partly

Reviewer #2: Yes

2. Has the statistical analysis been performed appropriately and rigorously? 

Reviewer #1: I Don't Know

Reviewer #2: I Don't Know

3. Have the authors made all data underlying the findings in their manuscript fully available?

Reviewer #1: Yes

Reviewer #2: Yes

4. Is the manuscript presented in an intelligible fashion and written in standard English?

Reviewer #1: Yes

Reviewer #2: Yes

5. Review Comments to the Author

Reviewer #1: I read the manuscript witch great interest ans I believe the HEPT training could be a nice first step in implementing palliative care in heart failure. Altough this article is about the improvement of a score the treating physisian experienced, it is not measuring the patients satisfaction and/or experiences, which is of course an very important outcome that realy makes a difference in daily practice. Have the authors any data on patient satisfaction (or ambition to investigate patients satisfaction/eperience?)

Another difficulty for me is the adjusment made to change the cancer PCPS and PCDS lists to heart failure by changing only "pain" to "symptom". By changing it it is no longer a validated tool, do the autors have any data which can validate this tool for use in heartfailure?

I appreciate the work in predicting which type of physian profits most from this course, but wouldn't it be nice just to implement sandard education of palliative care and ACP in heart failure in the education programme of every cardiologist? Please comment

Reviewer #2: I read with great interest this manuscript on physician education in terminal heart failure care. This is an underappreciated topic. Heart failure publications mainly focus on (novel) treatment modalities and rarely on the last phase of physican-patient interactions. From the reviewer's own perspective, this are difficult topics to discuss with the patient.

I have some comments to improve the manuscript:

1. the abstract reads very difficult, mainly due to the many abbrevations and it is difficult for the reader to distillate the main message.

2. The major limitation is a possible selection bias: Physicians that participate in the program are more likely to be motivated to improve their skills.

3. In most countries, heart failure clinics include specialized heart failure nurses or physician assistants that have more time for the patient and are crucial in the palliative phase. Please discuss.

4. The questionnaires are not validated for heart failure. Please discuss in the limitations and create a path towards validation.

5. It is very difficult to prove that the strategy actually improves patients' or families' well-being. I would suggest to set up a randomized trial aimed to tackle this question.

6. One of the important topics is deactivation of ICD therapy in palliative care of heart failure patients. Please discuss.

6. PLOS authors have the option to publish the peer review history of their article (what does this mean?). If published, this will include your full peer review and any attached files.

Reviewer #1: No

Reviewer #2: No

---

## [Author Response · Author response to Decision Letter 0]

18 Jan 2022

Responses to the Reviewer 1

Manuscript: PONE-D-21-27736/R1

Authors: Tatsuhiro Shibata, et al.

Title: Evaluation of the effectiveness of the physician education program on primary palliative care in heart failure

We thank Editor's specific comments and the Reviewer for her/his valuable comments. In line with the comments, we have revised our manuscript. Our detailed responses will follow the Editor and Reviewer’s comments. Our point-to-point responses are shown in the text in red to facilitate the review process.

Reviewer 1 comments:

I read the manuscript with great interest and I believe the HEPT training could be a nice first step in implementing palliative care in heart failure. Although this article is about the improvement of a score the treating physician experienced, it is not measuring the patients’ satisfaction and/or experiences, which is of course a very important outcome that really makes a difference in daily practice. Have the authors any data on patient satisfaction (or ambition to investigate patients satisfaction/experience?)

[Response]

Thank you very much for the valuable comment. We agree with the Reviewer. It is very important to measure the patients’ satisfaction and/or experience. Unfortunately, we do not have any data related to patients’ satisfaction/experience currently, which we should verify in the next study. Thus, we have added this issue in the Limitations section.

Page 22-23, lines 319-322. Further research investigating the impact on QOL and satisfaction of HF patients and their families will be needed to assess the true outcomes of primary palliative care education as our next study. Further, a randomized trial should be considered to evaluate the patients’ and/or families’ satisfaction.

[Comment]

Another difficulty for me is the adjustment made to change the cancer PCPS and PCDS lists to heart failure by changing only "pain" to "symptom". By changing it is no longer a validated tool, do the authors have any data which can validate this tool for use in heart failure?

[Response]

Thank you very much for your valuable comments. We fully agree with the Reviewer regarding this issue. We consider that it is also very important. However, there is no known tool to evaluate the effectiveness of education on palliative care for heart failure. Currently, previous studies have shown the strong similarities between issues related to palliative care for cancer and heart failure, as described below. 

Systematic review by Moens et al. (J Pain Symptom Manage. 2014;48(4):660-77.) has reported that there is a commonality of problems related to palliative care in heart failure compared to cancer. On the other hand, in a survey of palliative care for heart failure conducted by Kuragaichi et al. (Circ J . 2018;82(5):1336-1343.) at a cardiovascular teaching hospital in Japan, the most common symptom requiring palliative care for heart failure was dyspnea (91%), followed by anxiety (71%), depression (61%), and fatigue (57%), while pain (34%) was relatively rare.

In response to the issues above, we have changed the term to “symptom” instead of “pain”. However, we consider that it is required to validate this tool for heart failure. For the validation, we have to accumulate the data. According to the Reviewer’s comment, we have added this issue in the Limitations section.

Page 23-24, lines 331-334. Fourth, we used the tool, which has changed from “pain” to “symptom” of the cancer PCPS and PCDS list for heart failure. Currently, there is no tool to evaluate the effectiveness of education on HF palliative care. Therefore, we should validate this tool after data accumulation.

[Comment]

I appreciate the work in predicting which type of physician profits most from this course, but wouldn't it be nice just to implement standard education of palliative care and ACP in heart failure in the education programme of every cardiologist?

[Response]

I appreciate your comments. At present, palliative care is not included in the competency components of the training curriculum for cardiologists in Japan.

We hope that in future, palliative care will be included in pre- and post-graduate education in cardiology. We have added a description of the problems with this cardiovascular training curriculum in the “Discussion” section. 

Page 21, lines 285-288. However, palliative care is rarely included in undergraduate medical curricula, and is not included in the competency components of the current training curriculum for cardiologists in Japan. It should be included in pre- and post-graduate education in cardiology in near future.

Finally, we again would like to thank the Reviewer for the valuable comments on our work. We sincerely hope that our revised manuscript may again be considered for publication in the Journal. 

 

Responses to the Reviewer 2

Manuscript: PONE-D-21-27736/R1

Authors: Tatsuhiro Shibata, et al.

Title: Evaluation of the effectiveness of the physician education program on primary palliative care in heart failure

We thank Editor's specific comments and the Reviewer for her/his valuable comments. In line with the comments, we have revised our manuscript. Our detailed responses will follow the Editor and Reviewer’s comments. Our point-to-point responses are shown in the text in red to facilitate the review process.

Reviewer 2 comments:

I read with great interest this manuscript on physician education in terminal heart failure care. This is an underappreciated topic. Heart failure publications mainly focus on (novel) treatment modalities and rarely on the last phase of physician-patient interactions. From the reviewer's own perspective, this are difficult topics to discuss with the patient.

[Response]

Thank you very much for your favorable comments. 

[Comment 1]

The abstract reads very difficult, mainly due to the many abbreviations and it is difficult for the reader to distillate the main message.

[Response]

We apologize for this inconvenience. We have added abbreviation list (page 2, line 55-page 3, line 56) after the Abstract. Also, the full spelling of PT-HF was missing, which has been added to the Abstract.

Page 2, lines 36-39. Physician-reported practices, difficulties and knowledge were evaluated using the Palliative Care Self-Reported Practices Scale in HF (PCPS-HF), Palliative Care Difficulties Scale in HF (PCDS-HF), and Palliative care knowledge Test in HF (PT-HF), respectively.

[Comment 2]

The major limitation is a possible selection bias: Physicians that participate in the program are more likely to be motivated to improve their skills.

[Response]

Thank you for your comments. We agree with the Reviewer. We have enrolled all participants in this study; however, there may be a selection bias, because they have high motivation for HF palliative care. We have already raised this issue in the Limitations section, but added regarding the additional evaluation, which is needed after HEPT is widely adopted in the future.

Page 23, line 326-331. Third, selection bias may have affected the results. Physicians who participated in this study had a strong interest in palliative care for HF. They were relatively young, and 34.3% had a history of participation in PEACE. Further research is needed to determine whether the results apply to all types of physicians caring for patients with HF, and also to evaluate additionally if HEPT is widely adopted in future.

[Comment 3]

In most countries, heart failure clinics include specialized heart failure nurses or physician assistants that have more time for the patient and are crucial in the palliative phase. Please discuss.

[Response]

Thank you for your comments. Although this study was conducted for physicians, in clinical practice, non-physician medical staff such as heart failure nurses are also deeply involved in palliative care, and it is expected that a training system similar to HEPT will be established. This point has been added to the "Discussion" section. 

Page 22, line 311-314. Although this study was conducted for physicians, non-physician medical staff, such as heart failure nurses, are also deeply involved in palliative care in clinical practice. It is expected that a training system similar to HEPT will be established for medical staff.

[Comment 4]

The questionnaires are not validated for heart failure. Please discuss in the limitations and create a path towards validation.

[Response]

Thank you very much for your valuable comments. This is a very important issue. However, there is no known tool to evaluate the effectiveness of education on palliative care for heart failure. Currently, previous studies have shown the strong similarities between issues related to palliative care for cancer and heart failure, as described below. 

Systematic review by Moens et al. (J Pain Symptom Manage. 2014;48(4):660-77.) has reported that there is a commonality of problems related to palliative care in heart failure compared to cancer. On the other hand, in a survey of palliative care for heart failure conducted by Kuragaichi et al. (Circ J . 2018;82(5):1336-1343.) at a cardiovascular teaching hospital in Japan, the most common symptom requiring palliative care for heart failure was dyspnea (91%), followed by anxiety (71%), depression (61%), and fatigue (57%), while pain (34%) was relatively rare.

In response to the issues above, we have changed the term to “symptom” instead of “pain”. However, we consider that it is required to validate this tool for heart failure. For the validation, we have to accumulate the data. According to the Reviewer’s comment, we have added this issue in the Limitations section.

Page 23-24, lines 331-334. Fourth, we used the tool, which has changed from “pain” to “symptom” of the cancer PCPS and PCDS list for heart failure. Currently, there is no tool to evaluate the effectiveness of education on HF palliative care. Therefore, we should validate this tool after data accumulation.

[Comment 5]

It is very difficult to prove that the strategy actually improves patients' or families' well-being. I would suggest to set up a randomized trial aimed to tackle this question.

[Response]

Thank you for your comments. We agree with the Reviewer. It is very important to measure the patients’ satisfaction and/or experience. We will plan a randomized trial to evaluate the patients’ and/or families’ satisfaction. We have added this issue in the Limitations section.

Page 23, lines 319-322. Further research investigating the impact on QOL and satisfaction of HF patients and their families will be needed to assess the true outcomes of primary palliative care education as our next study. Further, a randomized trial should be considered to evaluate the patients’ and/or families’ satisfaction.

[Comment 6]

One of the important topics is deactivation of ICD therapy in palliative care of heart failure patients. Please discuss.

[Response]

Thank you for your comments. We agree that the topic on ICD deactivation is important.

ICD issues have been added to the Discussion section. 

Page 22, lines 305-310. In the clinical settings, the number of ICD implantations is increasing to prevent sudden death; however, shock therapy may be repeated at the end of life, resulting in patient distress, poor quality of death, and family distress. Currently, only a limited number of physicians have experience with ICD deactivation.12 We hope that the HEPT participants will discuss about ICD deactivation and lead to a wider dissemination of this concept.

Finally, we again would like to thank the Reviewer for the valuable comments on our work. We sincerely hope that our revised manuscript may again be considered for publication in the Journal.

---

## [Editor Report · Decision Letter 1]

21 Jan 2022

Evaluation of the effectiveness of the physician education program on primary palliative care in heart failure

PONE-D-21-27736R1

Dear Dr. Fukumoto,

We’re pleased to inform you that your manuscript has been judged scientifically suitable for publication and will be formally accepted for publication once it meets all outstanding technical requirements.

Kind regards,

Alexander H. Maass, M.D., Ph.D.

Academic Editor

PLOS ONE

Additional Editor Comments (optional):

The authors have significantly improved the manuscript.

---

## [Editor Report · Acceptance letter]

26 Jan 2022

PONE-D-21-27736R1 

Evaluation of the effectiveness of the physician education program on primary palliative care in heart failure 

Dear Dr. Fukumoto:

I'm pleased to inform you that your manuscript has been deemed suitable for publication in PLOS ONE. Congratulations! Your manuscript is now with our production department. 

Kind regards, 

on behalf of

Dr. Alexander H. Maass 

Academic Editor

PLOS ONE